# Dual Roles of Microglia in the Basal Ganglia in Parkinson’s Disease

**DOI:** 10.3390/ijms22083907

**Published:** 2021-04-09

**Authors:** Mohammed E. Choudhury, Yuka Kigami, Junya Tanaka

**Affiliations:** Department of Molecular and Cellular Physiology, Ehime University Graduate School of Medicine, Shitsukawa, Toon, Ehime 791-0295, Japan; mechoudhury81@gmail.com (M.E.C.); d401027z@mails.cc.ehime-u.ac.jp (Y.K.)

**Keywords:** synapse, phagocytosis, dopamine, basal ganglia, compensation, glutamate, subthalamic nucleus

## Abstract

With the increasing age of the population, the incidence of Parkinson’s disease (PD) has increased exponentially. The development of novel therapeutic interventions requires an understanding of the involvement of senescent brain cells in the pathogenesis of PD. In this review, we highlight the roles played by microglia in the basal ganglia in the pathophysiological processes of PD. In PD, dopaminergic (DAergic) neuronal degeneration in the substantia nigra pars compacta (SNc) activates the microglia, which then promote DAergic neuronal degeneration by releasing potentially neurotoxic factors, including nitric oxide, cytokines, and reactive oxygen species. On the other hand, microglia are also activated in the basal ganglia outputs (the substantia nigra pars reticulata and the globus pallidus) in response to excess glutamate released from hyperactive subthalamic nuclei-derived synapses. The activated microglia then eliminate the hyperactive glutamatergic synapses. Synapse elimination may be the mechanism underlying the compensation that masks the appearance of PD symptoms despite substantial DAergic neuronal loss. Microglial senescence may correlate with their enhanced neurotoxicity in the SNc and the reduced compensatory actions in the basal ganglia outputs. The dual roles of microglia in different basal ganglia regions make it difficult to develop interventions targeting microglia for PD treatment.

## 1. Introduction

Parkinson’s disease (PD) is a frequently diagnosed neurodegenerative disorder characterized by motor symptoms such as rigidity, bradykinesia, resting tremor, and postural instability, as well as nonmotor symptoms such as hyposmia, autonomic ataxia, and sleep disorders [1,2,3,4,5]. Pathologically, PD is characterized by the progressive degeneration of dopaminergic (DAergic) neurons in the substantia nigra (SN) pars compacta (SNc). The inclusion bodies in DAergic neurons, referred to as Lewy bodies, and their main protein component, α-synuclein (α-Syn), are the prominent neuropathological features of this disease [6]. The recently established PROPAG-AGEING project aims to characterize the contribution of the aging process to the development of PD [7]. A previous study showed that aged rats exhibited a higher susceptibility to neurotoxin 6-hydroxydopamine (6-OHDA) to produce a PD model than young rats [8]. In recent decades, the incidence of PD has increased rapidly with the increasing age of the population [9]. Besides conventional symptomatic treatments such as L-DOPA, there is a need to develop new interventions to treat and prevent PD while addressing its pathophysiology.

This review focuses on the involvement of microglia in PD pathogenesis. Microglia are one of the major types of glial cells and the only cell type of mesodermal origin in the central nervous system (CNS). They comprise 5–10% of all CNS cells [10,11] and are similar to macrophages and monocytes in the peripheral and circulatory systems [12,13], except that some markers make microglia and blood-borne macrophages distinguishable [14,15,16]. Microglia can release various neurotoxic mediators [17]. In the pathophysiological processes of PD, microglia play a role in DAergic neuronal degeneration in the SNc [18,19,20]. SNc DAergic neurons undergo degeneration at least partly due to genetic mutations [21,22,23,24] or the formation of Lewy bodies, which contain oligomerized insoluble α-Syn fibrils [6,25]. The degenerated neurons induce proinflammatory activation of microglia [26,27,28], leading to further degeneration of SNc DAergic neurons. A postmortem investigation of human brains showed that microglia transformed into reactive cells with proinflammatory phenotypes and expressed the major histocompatibility complex (MHC) class II antigen [19]. Thus, PD has neuroinflammatory disease characteristics, namely, the increased production of proinflammatory or neurotoxic factors by microglia in the SNc, such as nitric oxide (NO); reactive oxygen species (ROS); glutamate; chemokines; and cytokines such as CCL2, interleukin-1β (IL-1β), and tumor necrosis factor-α (TNF-α) (Table 1) [2,19,25,29,30,31]. All these microglia-derived substances can negatively affect SNc DAergic neuronal survival. For example, fibrillary aggregates of α-Syn enhance glutamate release by microglia, presumably resulting in the degeneration of DAergic neurons due to excitotoxic stress [32]. Microglia-activated astrocytes may mediate some of the neuroinflammatory reactions in PD pathogenesis [33,34,35].

In contrast, microglia also have neurotrophic or neuroprotective activities (Table 1) [36,37]. During inflammatory events, microglia can perform both proinflammatory and anti-inflammatory functions. Microglia exert their anti-inflammatory actions partly due to anti-inflammatory cytokines such as IL-10 and transforming growth factor-β (TGF-β) [38,39]. Microglia phagocytose degenerated cells and materials in brain tissues, likely suppressing proinflammatory responses by microglia and blood-borne leukocytes [40,41].

The motor symptoms of PD do not appear until most SNc DAergic neurons have degenerated and the striatal dopamine (DA) levels have reduced [2,5,64,65,66,67,68], although there is a pre-motor or prodromal stage during which nonmotor symptoms such as hyposmia or depression but not motor symptoms are apparent [5]. Some compensatory mechanisms delay the onset of PD motor symptoms, and many hypotheses regarding these mechanisms have been proposed. Microglia may contribute to these compensatory mechanisms by eliminating subthalamic nucleus (STN)-derived hyperactive glutamatergic synapses. Such synapses become hyperactive in response to the reduced striatal DA levels (Figure 1).

In this review, we discuss the functional differences between the microglia in the SNc and those in the basal ganglia outputs, SN pars reticulata (SNr), and globus pallidus pars interna (GPi) (Figure 1). Both regions project their GABAergic axons to both the thalamus and brain stem [69], and therefore, they are called basal ganglia outputs. SNc microglia may play a role in DAergic neuronal degeneration. Microglia in the SNc and basal ganglia outputs have distinct activation mechanisms. The microglia in the basal ganglia outputs may contribute to the compensatory mechanisms. Two neural circuits in the basal ganglia are of interest: the direct and indirect pathways (Figure 1A) [69,70,71]. The two circuits originate from two distinct GABAergic neuron populations in the striatum. In the direct pathway, GABAergic striatal neurons directly project their axons to the basal ganglia outputs. In the indirect pathway, another population of GABAergic neurons extend their axons to the globus pallidus pars externa (GPe). An optogenetic study clearly demonstrated that activation of the direct pathway enhances movement and that activation of the indirect pathway stops movement [71]. Because of their pro- and anti-inflammatory effects or their neurodestructive and neuroprotective roles, microglia are frequently described as “double-edged swords” [72,73,74,75]. However, in the present review, we use the term “dual roles” to refer to the distinct effects of microglia located in different regions of the basal ganglia on the pathophysiology of PD [68,76].

## 2. Microglia in the SNc of Brains with PD

### 2.1. Aggravating Actions of Microglia

Clinical and laboratory studies have revealed the presence of microglia with activated morphology with shortened processes and enlarged somata in the SNc of patients with PD. DAergic neuronal degeneration activates microglia in the SNc, which expresses and releases proinflammatory or neurotoxic mediators, ROS, glutamate, cytokines, and chemokines [12,35,45,46,76] (Table 1). Chemokines recruit leukocytes in the circulation, thereby aggravating neuronal degeneration [47,77]. Proinflammatory activation of microglia is often considered a secondary change following neuronal degeneration [12]. However, proinflammatory microglia negatively affect neuronal survival before DAergic neuronal degradation [78,79,80]. Furthermore, lipopolysaccharide (LPS) injection in and around the SNc of animals induces the proinflammatory activation of microglia and subsequent DAergic neuronal death [2]. The addition of LPS to a neuron–microglial cell coculture causes microglia to release significant amounts of NO, resulting in neuronal death. Suppressing the activity or expression of LPS-induced inducible NO synthase (iNOS) using a NOS inhibitor, glucocorticoids [12,81], noradrenaline (NA) [82], or bromovalerylurea [46] prevented neuronal death. These observations suggest that activated microglia cause neurodegeneration independently of the neuron-based neurodegeneration mechanisms.

IL-1β and TNF-α-expressing proinflammatory microglia may cause astrogliosis in the SNc, as shown in a 6-OHDA-induced PD model [18], although it remains unknown whether such astrogliosis protects DAergic neurons in the SNc. Even in PD animal models, astrocytes may play neuroprotective roles in the SNc by expressing ROS-scavenging enzymes and metallothionein [55] and by many other mechanisms [39,83,84,85,86]. However, neurotoxic astrocytes (called A1 astrocytes) were found to develop in the brains of patients with PD, presumably because activated microglia release IL-1α, TNF-α, and complement C1q [33]. IL-1α released from microglia may also stimulate aquaporin 4 expression in astrocytes, aggravating brain edema in ischemic brain insults [42]. Thus, activated microglia may enhance the degeneration of DAergic neurons in the SNc, partly by impeding such supportive actions of astrocytes for neurons.

The endogenous Toll-like receptor (TLR) ligands may trigger proinflammatory microglial activation. In a 1-methyl-4-phenyl-1,2,3,6-tetrahydropyridine (MPTP)-induced mouse PD model, TLR4-deficient mice were less vulnerable to MPTP intoxication than wild-type mice, and their SNc had fewer MHC class II antigen-expressing activated microglia [87]. Many other studies showed the involvement of TLRs in the activation of microglia in the SNc of PD-affected brains [88,89,90]. The endogenous TLR ligands in PD remain unknown. In ischemia- or injury-induced severe brain damage, damage-associated molecular patterns (DAMPs) released from degenerated neurons, other brain cells, and tissues act as endogenous TLR ligands [91,92]. TLR4 deficiency impairs the phagocytic response of microglia to insoluble intracellular inclusions formed in DAergic neurons or Lewy bodies containing α-Syn [93]. α-Syn activates proinflammatory microglia by inducing mitochondrial dysfunction or damage, which reduces ATP synthesis and leads to proinflammatory activation of the NLRP3 inflammasome [94]. Furthermore, α-Syn induces the microglial proinflammatory reactions partly by activating the signal transducer and activator of transcription 3 [95]. Thus, the mechanism of proinflammatory microglial activation by α-Syn in the SNc in PD may be distinct from that by DAMPs and other potential microglia activators such as neurons themselves [27], glutamate [68,96], or NA [97].

### 2.2. Favorable Actions of Microglia in the SNc

In PD, ROS play a role in SNc DAergic neuronal loss [98]. Activated microglia release ROS and NO [99]. However, microglia release fewer ROS than blood-borne macrophages [17]. In contrast, microglia markedly protect neurons from ROS-induced damage [36,37]. Microglia express neuroprotective factors such as IGF-1, platelet-derived growth factor (PDGF)-A, and hepatocyte growth factor (HGF) [12,46]. The neuroprotective properties of microglia are incompatible with their proinflammatory effects. The favorable phenotype may partly correspond to the so-called M2 phenotype of microglia. Microglia with proinflammatory properties have been called M1-polarized microglia, classically activated microglia, or simply M1 microglia. In contrast, microglia with neuroprotective, neurorestorative, and/or anti-inflammatory properties have been called M2 microglia or alternatively activated microglia [100]. However, researchers have questioned the M1 and M2 classification [38,41,101], partly because the expressions of markers used for this classification (such as IL-1β and CD86 for M1; and CD163 and IGF-1 for M2) are often inconsistent [12,41,46]. Microglia in the SNc of patients with PD may be regarded as M1-type cells with neurotoxic effects because they release proinflammatory mediators and neurotoxic substances. Furthermore, their phagocytic activity may affect damaged neurons. Microglia can phagocytose even viable damaged cells by recognizing the apoptosis marker phosphatidylserine on the outer leaflet of the plasma membrane of damaged cells [102]. Phagoptosis, a form of cell death by phagocytosis, can affect viable cells. It aggravates ischemic brain insults and may occur in PD [103].

### 2.3. Changes in Other Glial Cells

We investigated the effects of a cytokine mixture containing granulocyte/macrophage colony-stimulating factor (GM-CSF) and IL-3, both of which are hematopoietic cytokines that stimulate bone marrow cells. IL-3 increases the expression of the antiapoptotic factor Bcl-xL in neurons [61] and prevents ischemia-induced neuronal apoptosis. GM-CSF also suppresses the neurodegeneration of neurons in ischemic brain lesions [104]. The cytokine mixture of GM-CSF and IL-3 improved the morphological and functional outcomes of rats in a traumatic brain injury model [105], probably because it enhanced the proliferation of infiltrated blood-borne macrophages and increased the expression of their neurotrophic factors. The cytokines changed the morphology of rat primary microglia [106], increased their expression of the neuroprotective factors IGF-1 and HGF, and suppressed their expression of IL-1β and TNF-α [18] (Figure 2). Furthermore, the cytokines stimulated phagocytosis. When subcutaneously administered to rats with 6-OHDA-induced PD, the cytokine mixture inhibited DAergic neuronal loss, increased neuronal Bcl-xL expression, and suppressed microglial IL-1β expression in the SNc. SNc neurons and microglia both expressed cytokine receptors. These results indicate that activated microglia play pivotal roles in DAergic neuronal degeneration. The cytokine mixture creates an environment in which neurons can survive and the proinflammatory activation of microglia is avoidable (Figure 3).

In addition to well-known glial cells such as astrocytes, oligodendrocytes, and microglia, the brain parenchyma contains neural/glial antigen 2 (NG2) glia or oligodendrocyte progenitor cells. NG2 is a type of chondroitin sulfate proteoglycan with a molecular weight of 290–300 kDa [109]. Although the roles of NG2 glia have not been completely elucidated [108,110,111,112], distinct responses in NG2 glia and astrocytes were noted in the above cytokine study [18] (Figure 3). PD model rats who received the vehicle had a higher number of degenerated DAergic neurons and marked astrogliosis. In contrast, the SNc of rats who received the cytokine mixture had a markedly higher number of NG2 glial cells and exhibited almost no astrogliosis. NG2 expression, but not that of the microglial marker Iba1, correlated with DAergic neuronal survival [18]. These results suggest that the primary change in PD pathology is the decrease in neuronal viability and the secondary change is microglial activation. Astrogliosis occurs after neuronal loss. When the Bcl-xL expression is elevated by DAergic neurons and neuronal viability is improved, microglia activation favors NG2 glial proliferation. NG2 glia expressing the HGF receptor cMet proliferate in response to HGF released by microglia [113]. NG2 glia also express receptors for binding PDGF, and their survival is dependent on PDGF [108].

### 2.4. Interventions to Induce Favorable Actions of Microglia

Many studies have demonstrated that suppressing the proinflammatory activation of microglia leads to their neuroprotective phenotypes. Minocycline, a tetracycline antibiotic, prevents the proinflammatory activation of microglia [114,115,116]. In an MPTP-induced mouse model of PD, minocycline blockade of microglial activation showed neuroprotective effects [114]. Minocycline suppressed the expression of various proinflammatory mediators and enzymes, such as IL-1β, TNF-α, IL-6, cyclooxygenase-2, and iNOS. Furthermore, minocycline increased the expression of the anti-inflammatory cytokine IL-10 [116]. These effects are characteristic of minocycline, and other tetracyclines have much weaker effects on microglia. Based on the favorable effects of minocycline on activated microglia, clinical trials have attempted to determine whether it has ameliorative effects on PD. However, no satisfactory results have been obtained [117,118].

Glucocorticoids have strong immunosuppressive effects on microglia [12,46,81]. Dexamethasone (Dex), a synthetic glucocorticoid, prevents the release of NO by LPS-treated rat primary microglia, and it is much more efficient than minocycline [12]. Furthermore, Dex prevents the LPS-induced expression of IL-1β, TNF-α, and IL-6 and increases the expression of the neuroprotective factors IGF-1 and HGF. These results suggest that glucocorticoids induce M2-like phenotypes in microglia. Besides, in animal PD models, glucocorticoids, including Dex, can exert protective effects on SNc DAergic neurons and ameliorate motor deficits. Bromovalerylurea (BU), a type of hypnotic/sedative containing bromine, has a strong anti-inflammatory action [17,46,119,120]. Its immunosuppressive effect is as strong as that of Dex, and it increases the expression of IGF-1, PDGF, and HGF and suppresses the expression of IL-1β, TNF-α, iNOS, and the proinflammatory transcription factors called interferon regulatory factors. Oral administration of BU to 6-OHDA-induced PD model rats suppressed SNc DAergic neuronal degeneration and markedly ameliorated motor deficits [46]. The anti-inflammatory actions of BU may correlate with the suppression of glycolysis and mitochondrial metabolism, which reduce ATP production [17,119]. BU also suppresses LPS-induced phosphorylation of STAT1.

However, animal models produce acute neurodegeneration on an hourly or daily scale, which differs from the slow process of the neurodegeneration of clinical PD, which occurs on a yearly scale. The contribution of microglial inflammatory responses in clinical PD cases is still unclear. Although PD models are generally established in young animals, clinical PD usually affects the elderly, and researchers must consider the senescent changes in cells, as discussed next. Furthermore, activated microglia may have a role in the compensatory mechanisms (Figure 1), and Dex-induced microglial suppression may at least transiently aggravate the motor deficits [68]. Moreover, even if Dex, BU, and other anti-inflammatory agents and cytokines inhibit the neuroinflammatory processes, it would be difficult for patients with PD to continue taking the agents for decades because of their presumable side effects. These considerations highlight the difficulty in translating laboratory findings on anti-inflammatory drugs into clinical practice.

## 3. Microglia in the Basal Ganglia Outputs

### 3.1. Compensatory Mechanisms in PD

It is well-established that the severity of DAergic neuronal loss does not linearly correlate with the severity of PD motor symptoms, which become apparent only after a substantial loss of SNc DAergic neurons and the subsequent marked reduction in striatal DA levels. The estimated threshold for the degenerative changes is a 50–60% loss of SNc DAergic neurons and a 70–80% reduction in the striatal DA levels in humans as well as in PD animal models [2,64,65,121,122]. Two behavioral tests, the cylinder test and forepaw adjustment steps test, are used to evaluate the motor deficits in the forepaws of the rat hemi-PD model and are the most sensitive to decreases in the striatal DA levels. Yet, they detect motor deficits only when the DA levels are < 50% of the normal level. Thus, the compensatory mechanisms may suppress the appearance of motor symptoms.

Some compensatory mechanisms have been proposed to explain what prevents the decreased striatal DA levels from causing abnormalities in the neural circuits [64,65,122]. One of the probable compensatory mechanisms is the increased release of DA or the enhanced axonal sprouting by viable neurons. An increased DA turnover may also contribute to the compensation. Many other mechanisms, such as the involvement of enkephalin and increased sensitivity of D2 receptors, have been postulated. In relation to the increased sensitivity of dopamine receptors, apomorphine test should be noted, which is a common way to check the validity of the 6-OHDA-induced rat hemi-PD model. Apomorphine is a nonselective dopamine agonist used for the treatment of advanced PD [123], and it causes hemi-PD model rats to rotate instead of walking straight by overstimulating the hypersensitive dopamine receptor [2]. Microglia also play a part in the compensatory mechanisms (Figure 1).

### 3.2. Activated Microglia in the Basal Ganglia Outputs

In a previous study, while observing microglial activation in the brains of a rat PD model, we found that microglial activation was more prominent in the SNr and GPi than in the SNc (Figure 4) [68]. In PD, the basal ganglia outputs where GABAergic neurons localize have no apparent neuronal degeneration. The microglial activation was characterized by increased expression of CD11b immunoreactivity, a component of complement C3b receptor, and CD68, a phagocyte marker (Figure 4). Furthermore, the activated microglia characteristically expressed NG2, although it is a marker for NG2 glia. NG2 is often expressed by blood-borne macrophages as well as activated microglia, both of which possess increased phagocytic activity [39,40,109]. The NG2^+^ activated microglia adhered to the surface of GABAergic neurons in the basal ganglia outputs and engulfed synaptic elements. The engulfed synapses were STN-derived glutamatergic synapses located in the indirect pathway in the neural circuits in the basal ganglia (Figure 1). The decreased release of DA from SNc neurons hyperactivates STN. Microglia express various types of glutamate receptors [68,124,125,126], and glutamate can induce changes in their morphology [127] and activate their phagocytic functions [68,96]. Normally, none of the GABAergic neurons in the basal ganglia outputs are degenerated in PD brains; therefore, they do not directly affect the activities of microglia. The effects of glutamate neurotransmission may also be indirectly mediated to microglia by the increased extracellular concentration of ATP [127,128].

SNc DAergic neuronal loss activates striatal GABAergic neurons in the indirect pathway (Figure 1). The activated GABAergic neurons then inactivate GABAergic neurons in the GPe and hyperactivate glutamatergic neurons in the STN. Excessive STN neuron activity causes glutamate spillover from the hyperactive glutamatergic synapses in the basal ganglia outputs, which stimulates the phagocytic activity of microglia [68,96]. In turn, microglia eliminate the hyperactive glutamatergic synapses (Figure 1B). Thus, activated microglia in the basal ganglia outputs at least partially normalize the abnormality in the neural circuits in the basal ganglia in PD. This is another probable compensatory mechanism involving microglia.

Functional neurosurgery treats PD using deep brain stimulation (DBS) to suppress the activity of the hyperactive STN. STN-DBS reduces motor symptoms in patients with PD and improves their quality of life, making STN the current primary target of DBS [129,130,131]. The effects of activated microglia in the basal ganglia outputs may be similar to those caused by STN-DBS. During the presymptomatic stages of PD, when DAergic neuronal degeneration progresses inconspicuously, microglia may prevent the appearance of PD symptoms by eliminating hyperactive STN-derived synapses (Figure 1B and Figure 4C). Why, then, is neurosurgical STN-DBS effective in patients with PD despite the DBS-like effects of microglia? We speculate that microglial senescence causes the appearance of PD symptoms in the elderly (Figure 1C).

### 3.3. Problems Associated with Anti-Inflammatory Drugs That Suppress Microglial Phagocytosis

Chronic administration of a synthetic glucocorticoid such as Dex or BU effectively suppresses microglia-mediated DAergic neuronal degeneration in animal PD models. In fact, after ceasing administration, the PD model rats displayed better motor functions than the control rats [12,46]. However, we found that a high dose of Dex transiently aggravated the motor symptoms of rats with 6-OHDA-induced PD, even if it ameliorated the overall outcome [68]. Besides its anti-inflammatory effects, Dex strongly suppresses the phagocytic activity of primary cultured rat microglia and CD68 expression by microglia in the SNr of PD model rats. These observations suggest that microglia have dual roles in PD pathophysiology (Figure 1): aggravating effects in the SNc through their proinflammatory actions, thereby enhancing DAergic neuronal degeneration, and favorable effects in the basal ganglia outputs by normalizing the glutamatergic neurotransmission through the phagocytosis of the hyperactive synapses. Thus, pharmacological interventions that suppress microglial activities may not succeed for two reasons. First, because after most neurons have degenerated, it is too late to start a clinical intervention. Second, because they prevent the favorable phagocytic actions of microglia in the basal ganglia outputs. Stress conditions increase the release of glucocorticoids from the adrenal glands [11]. Emotional or psychological stress can transiently worsen the motor symptoms of patients [132] and animals [133]. The stress-induced aggravation of motor symptoms at least partly correlates with the elevation of circulating glucocorticoid levels, which impairs the compensatory actions of microglia in the basal ganglia outputs.

In contrast, the cytokines IL-3 and GM-CSF increase microglial phagocytic activity (Figure 2), whereas they suppressed the proinflammatory activation of microglia in the SNc in a rat PD model [18]. Although chronically administering the cytokine mixture to patients with PD may be impossible, this is how an ideal therapeutic intervention should affect microglia.

## 4. Senescent Changes of Microglia and PD

Aging induces profound changes in microglial morphology and functions [134]. In the mouse brain, aged or senescent microglia have characteristic short, thick processes and enlarged cell bodies, as well as showing increased production of inflammatory cytokines and ROS and impaired phagocytic and lysosomal activity. Overall, senescent changes promote microglial neurotoxicity. Similarly, in the human brain, aged dystrophic microglia display spherical swelling and reduced arborization of processes [135], endoplasmic reticulum dilatation, and an abundance of lipofuscin deposits [136,137]. Transcriptomic analysis of sorted microglia in aged human brains showed a high expression of cell adhesion, axonal guidance, and cell surface receptor genes [138]. Increased expression of MHC class II [139,140], CD68 [136,141], and TLRs [142] are common features of aged microglia. A study reported a subset of microglia with inflammatory phenotypic signatures and high CD11c and CD14 in aged mice [143]. The senescent changes of microglia cause “inflammaging”, a chronic, low-grade inflammation regarded as the most common brain feature in aged individuals [144]. From neuroprotective cells, aged microglia become neurotoxic cells that cause inflammaging [145]. The phenotypic changes of aging microglia may correlate with the PD symptom onset by inducing SNc DAergic neuronal degeneration and also by reduced phagocytic elimination of hyperactive glutamatergic synapses from STN (Figure 1).

Besides the functional features of individual microglia, aging affects the number of microglia. An immunohistochemical study on aged mouse brains showed that the number of Iba1^+^ microglia was drastically decreased and that their distribution pattern was uneven and unbranched [146]. We also observed microglial phenotype changes in the aged rat prefrontal cortex using flow cytometry (Figure 5). Microglia from 22-month-old rats had higher forward scatter (FS; cell size index) and side scatter (SS; granularity index) values than those from 2-month-old rats. Therefore, microglia in aged brains are larger and retain higher contents of intracellular phagosomes or lysosomes than those in young brains. Figure 5 also shows the increased expression of the phagocytosis-related molecules CD11b, CD68, and NG2, as well as proinflammatory phenotype-related markers CD86 and CD45 in aged microglia [147]. Nevertheless, the overall phagocytic activity of aged microglia declines, and a reduced ability to eliminate hyperactive synapses in the basal ganglia outputs may allow the motor disorders of PD to appear [148].

Similar to DAergic neurons, noradrenergic neurons in the locus coeruleus degenerate with age [149]. Although there is much debate [97], NA inhibits microglial proinflammatory activation [82,150] at least partly by increasing intracellular cAMP levels [150,151]. In Alzheimer’s disease, activated microglia accelerate neuronal loss [152]. Microglial activation may correlate with the decrease in brain NA levels due to the senescent changes of locus coeruleus neurons [149]. Furthermore, aging gradually increases the number of degenerating neurons, resulting in a massive loss of CD200 or CX3CL1, which inhibits the proinflammatory activation of adjacent microglia [153].

## 5. Conclusions and Prospects

Many investigations have focused on the pathophysiological contribution of SNc microglia to PD. Therefore, microglia are recognized as promoters of DAergic neuronal death, and researchers have attempted to use anti-inflammatory agents for PD treatment. However, as introduced in detail in this review, microglia may play critical homeostatic roles to maintain motor functions by eliminating STN-derived hyperactive glutamatergic synapses. Synaptic elimination by microglia through phagocytosis occurs in the developing brain and the normal mature brain [154,155,156]. A deficiency in synaptic elimination by microglia may cause developmental behavioral disorders such as autism spectrum disorder [76]. A deficiency in the mature brain may correlate with sleep/wake cycle disorders [96], memory function disorders [157], and psychiatric or stress-related mental disorders [76,158,159]. The most significant role of microglia in the mature brain with almost intact blood–brain barrier (BBB) functions might be the maintenance of neural circuits through synapse phagocytosis. BBB disruption allows the massive infiltration of various types of factors, including cytokines and amino acids [68,96,106,107], resulting in the proliferation and proinflammatory activation of microglia. Yet, such apparent BBB breakdown does not occur in PD, and DAergic neuronal degeneration progresses slowly. The proinflammatory activation of SNc microglia in PD is less pronounced than that in stroke [39] or brain injury [17,160]. Microglial activation is stronger in the SNr or GPi than in the SNc, suggesting that microglia prevent PD onset rather than worsening PD. Progressive DAergic neuronal degeneration is an inevitable physiological change that occurs with aging, and it seems counterintuitive to assume that microglia primarily aggravate neuronal and neural circuit damage. A greater focus on the favorable effects of microglia on homeostasis in the brain functions would help develop new therapeutic avenues for PD.

## Figures and Tables

**Figure 1 ijms-22-03907-f001:**
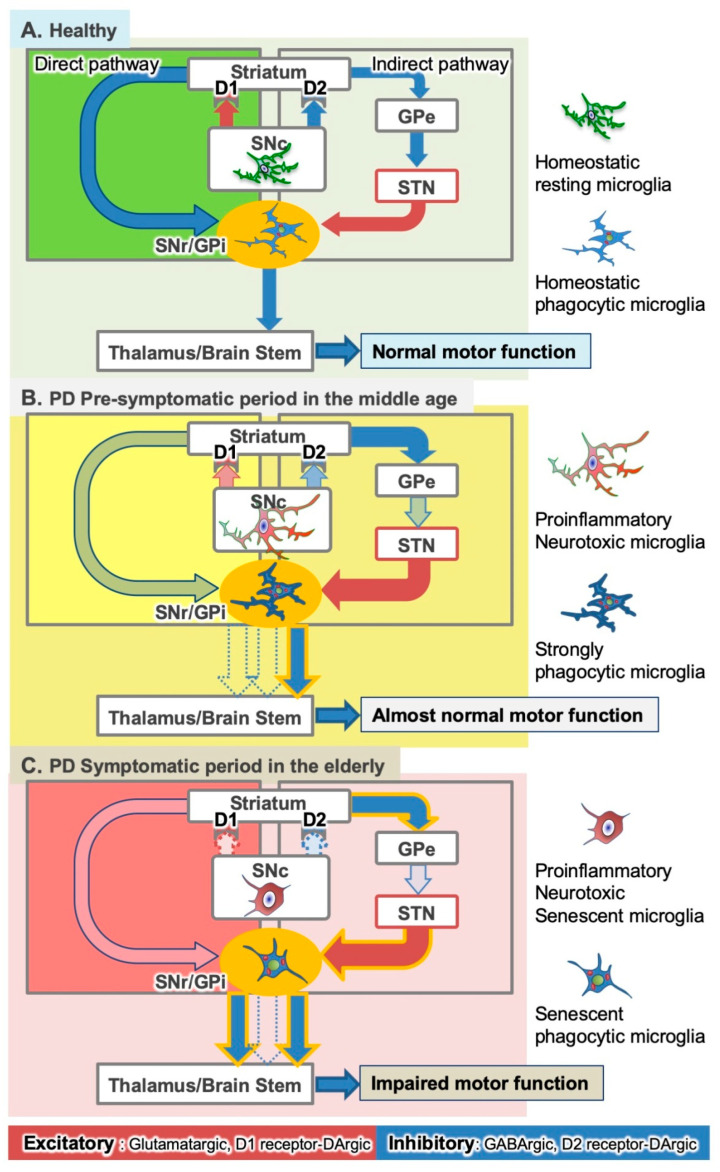
Hypothetical dual roles of microglia in the pathophysiology of Parkinson’s disease (PD). (**A**) In the basal ganglia, homeostatic (or ramified) microglia may play a role in the maintenance of neural circuits both in the substantia nigra (SN) pars compacta (SNc) and basal ganglia outputs (SN pars reticulata (SNr) and globus pallidus pars interna (GPi)) in the normal young adult brain. STN; subthalamic nucleus, GPe; globus pallidus pars externa. (**B**) With age, SNc dopaminergic (DAergic) neuronal degeneration progresses slowly, and microglia undergo gradual proinflammatory activation. In the indirect pathway, DAergic neuronal degradation hyperactivates glutamatergic STN neurons. However, microglia in the basal ganglia outputs prevent the development of PD symptoms by eliminating hyperactive glutamatergic synapses from the STN. (**C**) Finally, most DAergic neurons die, and motor deficits appear. Microglia undergo senescent changes. In the SNc, microglia still exert harmful effects on viable DAergic neurons. In the basal ganglia outputs, senescent microglia cannot normalize the hyperactive synapses.

**Figure 2 ijms-22-03907-f002:**
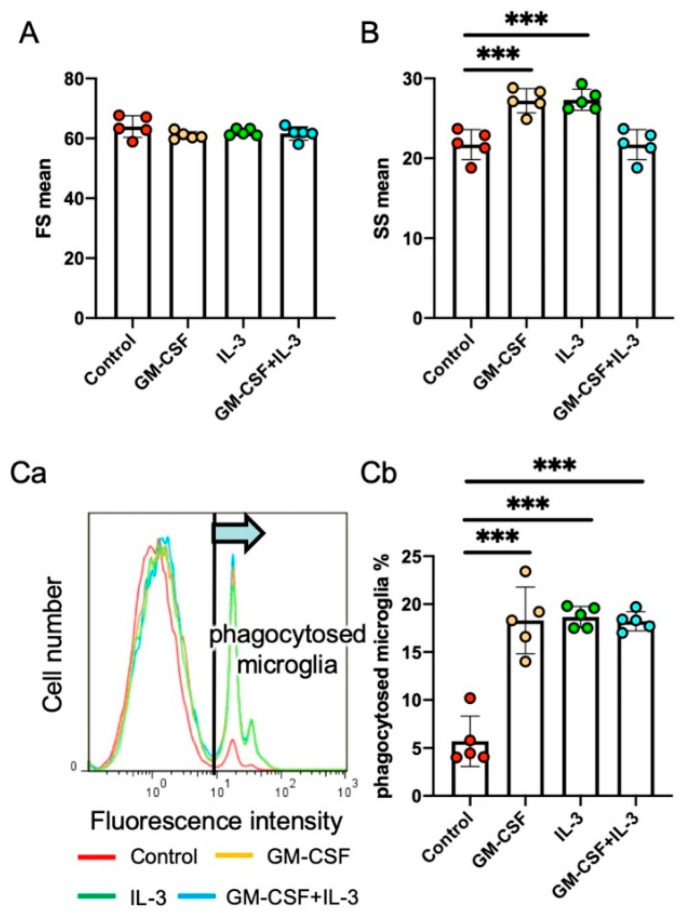
Effects of the cytokines granulocyte/macrophage colony-stimulating factor (GM-CSF) and IL-3 on primary rat microglia. Effect of GM-CSF, IL-3, and GM-CSF + IL-3 on the morphology of rat primary microglia (forward scatter (FS; [**A**]) and side scatter (SS; [**B**])) and their phagocytic internalization of fluorescence-labeled latex beads (C, representative histogram (**Ca**) and the percentage of phagocytosed microglia (**Cb**)) as measured by flow cytometry. Data are expressed as mean ± standard deviation. Unpaired two-tailed t-test. *** *p* < 0.001. Detailed information on the methodology for preparation of rat primary microglia culture and flow cytometry analyses is described elsewhere [96,107].

**Figure 3 ijms-22-03907-f003:**
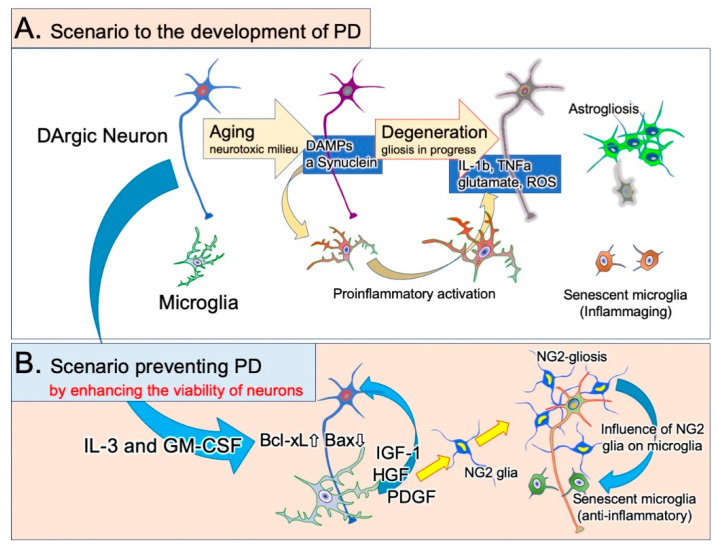
Neuronal viability and microglial phenotype changes in the SNc of a rat PD model. (**A**) With age, a toxic milieu forms around the DAergic neurons and adds to the intrinsic toxic mechanisms. Degenerating neurons generate damage-associated molecular patterns (DAMPs) and accumulate Lewy bodies containing α-Syn fibrils within their somata, inducing proinflammatory microglia. Activated microglia further stimulate neuronal degeneration by releasing neurotoxic mediators. Finally, most DAergic neurons are lost and astrogliosis occurs. Senescent microglia may lose their neurotrophic properties but can still generate neurotoxic mediators. (**B**) The cytokine mixture (GM-CSF + IL-3) can inhibit progressive neurodegeneration by increasing the expression of the antiapoptotic factor Bcl-xL and suppressing the expression of the proapoptotic factor Bax. Then, microglial neurotoxic activation is avoided and activated microglia release more neurotrophic factors. Microglia-derived growth factors play neuroprotective roles and enhance the survival of DAergic neurons. Microglia-derived PDGF and HGF stimulate the proliferation of neural/glial antigen 2 (NG2) glia, resulting in NG2 gliosis. NG2 glia may be involved in maintaining the homeostatic functions of microglia. More detailed information is described elsewhere [18,108].

**Figure 4 ijms-22-03907-f004:**
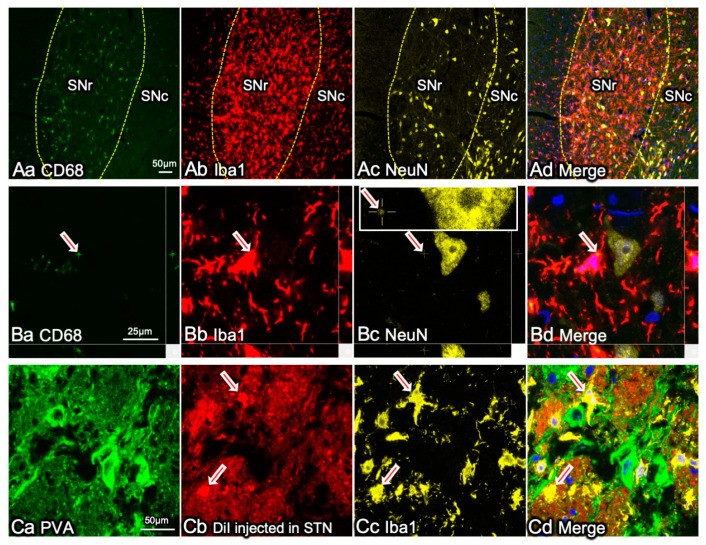
Microglial activation is more pronounced in SNr and GPi than in SNc. (**A**) Activated microglia with elevated CD68 expression are more densely present in SNr rather than in SNc in a rat 6-OHDA-induced PD model brain. The cryosection was triple-immunostained with antibodies to a phagosome marker CD68 (**Aa**), a microglial marker Iba1 (**Ab**), and a neuronal marker NeuN (**Ac**). (**B**) An activated microglial cell in SNr has a CD68^+^ phagosome (arrows) in its cytoplasm that contains NeuN^+^ materials (an arrow in inset in (**Bc**)), indicating that the cell phagocytosed some neuronal elements. (**C**) Activated microglia in the GPi phagocytosed nerve terminals from the STN. Lipophilic red fluorescent DiI was injected into the STN. The DiI was transported on the axonal membrane to the GPi. Activated Iba1^+^ microglial cells internalized DiI (arrowheads in **Cb**, **Cc**, and **Cd**). GABAergic neurons are identified with immunoreactivity to parvalbumin (PVA; **Ca**). The methodology for these results is described elsewhere [68].

**Figure 5 ijms-22-03907-f005:**
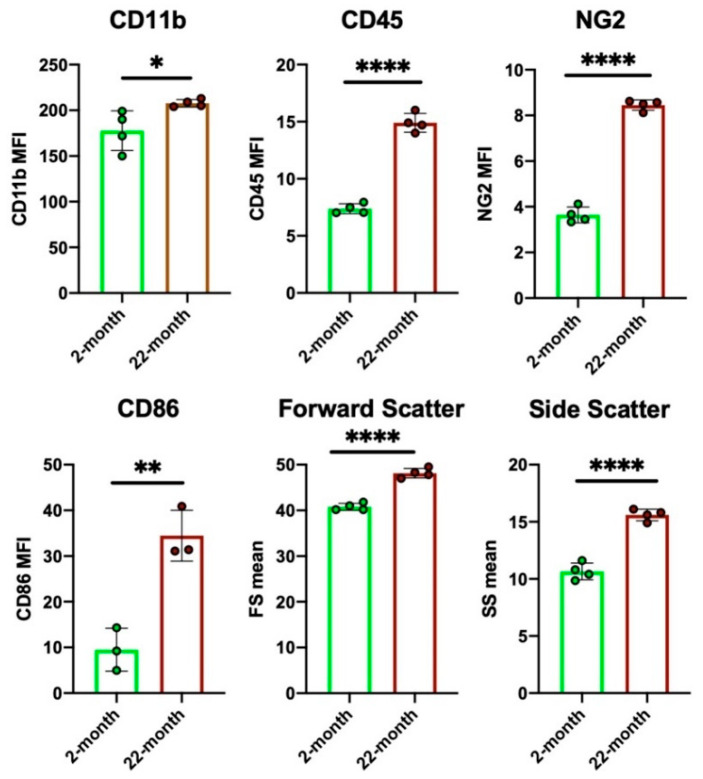
Senescent changes of microglia, as revealed by flow cytometry. Brain cells in the prefrontal cortices of 2- and 22-month-old male Wistar rats were enzymatically dissociated and subjected to flow cytometry analyses. Microglial cells in the 22-month-old rat brains showed higher CD11b, CD45, NG2, and CD86 expression than those in the 2-month-old rat brains. Furthermore, the senescent microglia had higher forward scatter (FS) and side scatter (SS) values, implying that they had larger and more granular somata than those in the young rat brains. Data are expressed as the mean ± SD. Unpaired two-tailed t-test. * *p* < 0.05, ** *p* < 0.01, **** *p* < 0.0001. Detailed information on the methodology is presented elsewhere [96].

**Table 1 ijms-22-03907-t001:** A list of neurotoxic and neuroprotective factors released by microglia.

**Neurotoxic and/or Proinflammatory Factors**	**Roles**
Interleukin-1β (IL-1β)	typical pro-inflammatory cytokine [42,43]
Tumor necrosis factor-α (TNF-α)	typical pro-inflammatory cytokine [42,43]
IL-1α	typical pro-inflammatory cytokine, modulating astrocyte properties [42,44]
Glutamate	excitotoxin inducing neuronal death [45]
Inducible nitric oxide synthases (iNOS) or NO	causing neuronal apoptosis [36,46]
Reactive oxygen species (ROS)	causing DNA damage and apoptosis [17]
CCL2	recruitment of circulating leukocytes [47]
IL-5	increasing nitrite levels [48]
IL-8	inducing release of pro-inflammatory cytokines and COX-2 [49]
IL-12	inducing nitric oxide synthase and activation of NFκB [50]
IL-15	inducing release of nitric oxide [51]
IL-18	inducing release of pro-inflammatory cytokines [52]
Cyclooxygenase 2 (COX 2)	mediating microglial activation and neurodegeneration [53]
**Neuroprotective and/or anti-inflammatory factors**	**Roles**
insulin-like growth factor-1 (IGF-1)	inhibiting microglial ROS generation and suppressing M1 phenotype [17]
platelet derived growth factor (PDGF)	inhibiting neuronal apoptosis PDGF [54]
hepatocyte growth factor (HGF)	supporting regeneration of damaged neuron [55,56]
transforming growth factor-β (TGF-β)	inhibiting activation of NFκB [38]
brain derived growth factor (BDNF)	inhibiting the pro-inflammatory activation [57]
arginase 1 (Arg-1)	inhibiting nitric oxide release [58]
glial cell-derived neurotrophic factor (GDNF)	increasing enzymatic activity of superoxide dismutase [59]
manganese-dependent superoxide dismutase (MnSOD)	suppressing oxidative stress [60]
IL-3	inhibiting apoptotic neuron death [61]
IL-4	increasing expression and release of IGF-1 [62]
IL-10	suppressing caspase-1-dependent IL-1β maturation [63]

## Data Availability

Not applicable.

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
