# Peer review of "Dual Roles of Microglia in the Basal Ganglia in Parkinson’s Disease"

_ijms, 2021, doi:10.3390/ijms22083907_

Round 1
Reviewer 1 Report
I think this is a very interesting review about the dual effects of activated microglia on PD.
As a suggestion, I would like you to add a table as a summary of the effect of different substances (like glutamate, etc) on the microglia and indicating if anti- or pro-inflammatory effect is triggered.
Do you have any image of immunohistochemical staining in GPi to complement figure 3?
Author Response
Please see it in the attached file.
Reviewer 2 Report
The present review is focused on understanding the roles od microglia in the pathophysiology of Parkinson’s disease (PD). Even though it is a major mechanism mediating non-cell autonomous mechanisms mediated dopaminergic neurodegeneration and the authors present relevant information on the changes of microglial phenotypes in different areas of the brain, the manuscript lacks a clear message demonstrating these evidences. Therefore, major revisions should be made to improve its clarity and organization:
- The abstract lacks a clear identification of the main aim of the review; it correctly addresses the main implications of microglia activation and senescence to PD, but does not identify the major contribution of this review to the field;
- Lines 30-32: in PD, the inclusion of Lewy bodies is not exclusive to dopaminergic neurons;
- Line 47: genetic variation itself is not an intrinsic mechanism of SNc dopaminergic neurons degeneration; it would be relevant to identify the genes and mutation variants most commonly associated to PD (e.g. LRRK2- G2019S), as well as identify their influence on glial cells;
- Line 65: although PD cardinal motor symptoms are only more pronounced when most of SNc dopaminergic neurons are lost, there are some subtle motor alterations and robust evidence of the existence of non-motor symptoms before major SNc degeneration;
- Line 71: although the authors identify the discussion of functional differences between microglia in the SNc and basal ganglia outputs as the main focus of the review, this topic only appears as one section in the review
General comments
- Some references are missing;
- Loose sentences, no connection between sentences and paragraphs;
- No development of important topics, such as :
- Section 3.1: what is the link between animal behavior (apomorphine induced rotation behavior and cylinder test) and the compensatory mechanisms in PD? what is the relevance of microglia as a compensatory mechanism?
- Section 3.2: why is there no evidence of microglia activation in other outputs of the basal ganglia in PD patients? what’s the relevance in identifying this activation in the PD animal model?
- Confusing order of the sections presented in the manuscript. Probably it would be more interesting if the authors describe the senescent changes of microglia in PD in the beginning of the manuscript.
- Conclusion presents new information, which would be relevant to have been discussed in the previous sections of the manuscript
Author Response
Please see it in the attached file.

Reviewer 3 Report
- This manuscript still need a complete proofreading in English grammar.
- The author' discussion only focused on microglia in the basal ganglia so the 'big' title should be revised.
- There are 2 Figure4 in the manuscript. The Figure 3 is too simple. The author should arrange the figures according to the role of the journal.
- As a review, the author should not publish their own experimental data. If some data are essential for the discussion, the author should supply 'Materials and Methods' section.
- The Figure4 is the central scheme of the review and only was cited 2 times in the text. The author should emphasize the figure.
Author Response
Please see it in the attached file.

Round 2
Reviewer 2 Report
The revised version provided by the authors has been improved and the authors’ responses are acceptable. Nevertheless, there are still some concerns with the paper in its current form:
1) there is a repetition of the section 2.4 in section 3, with the addition of a new paragraph;
2) my issue with section 3.1 is not in understanding the role of microglia as a compensatory mechanism, but mainly the connection between apomorphine induced rotation behavior and this possible role of microglia. Apomorphine is a dopamine receptor agonist that induces an hyperstimulation of the supersensitive dopamine receptors in the lesioned striatum, therefore its effects on unilaterally-lesioned animals rotation will only be visible if there is a significant nigrostriatal degeneration (>90%). Otherwise, the stimulation of still active (not hypersensitive) dopamine receptors in the lesioned striatum, will not induce a rotation behavior. Moreover, amphetamine can be used to identify partially lesioned animals, as it induces a rotational behavior with as little as 50% cell loss. Therefore, in this case I would argue that the lack of apomorphine induced rotational behavior in partially lesioned animals is a “methodological” feature, rather than the result of a microglia compensatory mechanism at basal ganglia output.
3) regarding section 3.2, my question is related with the translational significance of this possible role of microglial compensatory mechanism in basal ganglia outputs. Is there any evidence for microglial activation in SNr and GPi in PD patients’ brains?
Reviewer 3 Report
- The author supplied some useful informations (eg. Table 1) and improved the manuscript.
- The author can add some research data in the review, but should not state too much informations of 'Materials and Methods' and 'Results' in figure legends. Figure legends should only precisely state major figure data.
- The author should supply a 'Materials and Methods' as a supplimentary file, and move the 'Resultes' informations from Figure legends to main text.
- Review is a review, but unpublished data must be strictly illustrated according to research article format. If not, the reader can not believe the data and repeat it.
